# ESG Performance and Enterprise Value in China: A Novel Approach via a Regulated Intermediary Model

**Xuming Shangguan [1,2], Gengyan Shi [3] and Zhou Yu [4,***

1   Business School, Xinyang Normal University, Xinyang 464031, China; tonyshop@126.com
2   Dabie Mountain Economic and Social Development Research Center, Xinyang Normal University, Xinyang 464031, China
3   College of Business Administration, Capital University of Economics and Business, Beijing 100070, China; yan6628@foxmail.com
4   Department of Family and Consumer Studies and Asia Center, University of Utah, Salt Lake City, UT 84112, USA
*   Correspondence: zhou.yu@utah.edu

**Abstract:** ESG (environmental, social, and governance) performance increasingly influences enterprise valuation. While researchers debate about the precise nature of this influence, most assume a positive linear relationship. This study introduces a novel ESG responsibility performance metric utilizing a regulated intermediary model using representative data synthesized from leading ESG rating agencies in China. It investigates the pathways of this influence and examines the mediating effects of corporate reputation, stakeholder engagement, and regulatory compliance. The findings reveal an inverted U-shaped relationship between ESG performance and enterprise value, moderated significantly by financing constraints. These findings remain robust after accounting for potential endogeneity using instrumental variables. Heterogeneity analysis highlights that this inverted U-shaped relationship depends on the industry characteristics and ownership structures, particularly noticeable in non-polluting and non-state-owned enterprises. Moreover, enhanced ESG performance correlates with a reduced cost of equity financing, thereby augmenting enterprise value. Financial institutions might consider employing innovative financial instruments to diversify their enterprise financing channels and effectively bolster ESG-focused enterprises.

**Keywords:** ESG; enterprise valuation; corporate sustainability; the regulated intermediary model

## 1. Introduction

Environmental, social, and governance (ESG) principles underscore that enterprises ought to incorporate environmental and social considerations into their business practices. For ESG responsibility fulfillment, enterprise behavior is assessed by examining the interplay of environmental, social, and governance performance (Li et al., 2021; Eccles and Stroehle, 2018) [1,2].

From 2016 to 2020, global ESG-related investments witnessed remarkable growth, expanding sevenfold with an annualized growth rate of 63% [3]. While ESG practices in Chinese enterprises initially lagged behind their European and North American counterparts, they have gained significant momentum in recent years.

In September 2020, the Chinese government introduced an ambitious climate target known as the "Dual Carbon Goal" [4], aiming to achieve two objectives: carbon peaking by 2030 and carbon neutrality by 2060. The 20th CPC National Congress reaffirmed this target in 2022, urging enterprises to actively participate in global environmental governance.

By the end of 2021, over half of all Chinese enterprises had released ESG performance reports, as reported by the China Listed Companies Association. Fulfilling ESG responsibilities has become an imperative for businesses in China (Li et al., 2023) [5].

Despite the rapid growth of ESG practices, there are conflicting research findings regarding their impact on enterprise value. On the one hand, neoclassical theory posits that private enterprises must prioritize profit and maximize returns for shareholders. Fulfilling ESG responsibilities can be detrimental to enterprise value due to resource diversion, upfront costs, and reduced short-term profits (Wieczorek et al., 2021; Gillan et al., 2021) [6,7].

Kim and Lyon (2015) contend that some companies fulfill their ESG responsibilities mainly to avoid regulatory penalties, resulting in burdens without direct value creation [8]. Sassen et al. (2016) find negative impacts on the value of European companies [9]. Brammer and Pavelin (2010) also note voluntary fulfillment of ESG initiatives may decrease enterprise value if it is perceived as insincere by investors, hindering trust and ultimately undermining overall worth [10].

On the other hand, Porter (2006) views ESG responsibilities as opportunities for innovation and competitive advantage, which can enhance enterprise value [11]. Aligned with stakeholder theory, fulfilling ESG responsibility promotes a positive corporate reputation, investor relations (Huang, 2019) [12], and effective communication and engagement (Branco and Rodrigues, 2006; Revere, 2009) [13,14]; lowers creditors' required rate of return (Huang et al., 2022) [15]; and improves management capabilities (Xie, 2014; Islam et al., 2021) [16,17].

Friede et al. (2015) found that ESG responsibility fulfillment contributes to increasing corporate value in 90% of studied cases [18]. Xu et al. (2022) argue that ESG fulfillment through information disclosure enhances transparency, investor support, and capital market value [19]. Similarly, Yi et al. (2022) show that fulfilling ESG responsibilities has a positive impact on corporate value [20].

Considering these conflicting results, more research is needed to uncover the precise impact pathways, especially in China, where regulatory compliance holds unique importance. This paper aims to delve deeper into the motivations behind ESG responsibility fulfillment, including meeting regulations (Lokuwaduge and Heenetigala, 2017) [21], enhancing reputation and competitive advantages (Jasni et al., 2020) [22], and strengthening stakeholder relationships (Li et al., 2022) [23].

Chinese listed enterprises often encounter special challenges in securing funding, leading to a financing gap and higher financing costs (Leitner, 2016) [24]. These elevated financing constraints can compel enterprises to forego profitable opportunities, resulting in resource loss and hindering enterprise value (Ma, 2019) [25]. Chen and Yu (2022) [26] found that financing constraints have hampered ESG performance, damaging enterprise value. Similarly, Wang et al. (2022) [27] suggest that financing constraints have mediated the relationship between ESG performance and enterprise value.

In addition, several studies demonstrate a negative association between ESG performance and enterprise financing costs (Hamrouni et al., 2019; Raimo et al., 2021; Feng and Wu, 2021; Gigante and Manglaviti, 2022; Chouaibi et al., 2021) [28–32]. Strong ESG performance can attract investor attention, potentially reducing financing costs (Mansouri and Momtaz, 2022) [33].

The connection between financing costs and enterprise value is well established (Chen et al., 2010) [34]. Liu (2020) [35] highlights that high financing costs hinder a company's growth. Recent research has investigated the interplay between ESG, financing costs, and enterprise value (Henisz et al., 2019) [36]. Wang and Yang (2022) [27] show that strong ESG performance improves enterprise value by lowering financing costs. Feng and Wu (2021) demonstrate that companies with a history of strong ESG performance are more likely to secure funding during a crisis, such as the COVID-19 pandemic, exhibiting higher enterprise value. In summary, the existing research generally suggests that a Chinese company's ESG performance is positively associated with its enterprise value in lowering its financing costs.

Based on the existing research, this paper identifies three key limitations and provides corresponding solutions: (1) Previous empirical research has relied on inconsistent ESG performance indicators collected by different institutions, resulting in potentially unreliable

and incomparable research findings. To address this problem, this study synthesizes data from prominent ESG rating agencies in China and constructs a comprehensive indicator for ESG performance. Because of the growing popularity of ESG investing, numerous third-party ESG evaluation agencies have emerged recently and made this possible.

(2) The existing empirical studies have assumed a linear relationship between ESG performance and enterprise value, leading to incongruent results across studies. To overcome this limitation, this paper adopts a specialized regression model, generating a cohesive empirical framework for future research.

(3) Prior studies often overlook the potential problem of biased estimation due to endogeneity. To tackle this issue and reduce threats to internal validity, this paper employs instrumental variables via the two-stage least-squares method. Specifically, after rigorously testing instrumental variables such as analyst prediction bias and the industry's mean ESG responsibility performance, we will incorporate them into our analysis.

The remainder of the paper proceeds as follows: Section 2 reviews the literature and outlines hypotheses regarding the impact of ESG responsibility fulfillment on enterprise value. Section 3 develops a regulated intermediary model and presents a framework for examining the complex relationship between ESG performance and enterprise value in China. Section 4 presents the main results. Section 5 concludes.

## 2. Theoretical Analysis and Hypotheses Development

### 2.1. Nonlinear Relationships and the Moderating Effect of Financing Constraints

It is widely acknowledged that enterprises fulfill their ESG responsibilities to meet regulatory requirements, investor demand, and stakeholder expectations. Meanwhile, scholars have extensively researched methods for maximizing enterprise value. According to stakeholder theory, enterprises generate value by consistently engaging with stakeholders, with stronger connections enhancing value. First, ESG performance builds a positive brand image, attracting customers (Wang and Xu, 2016) [37]. Second, it fosters effective communication with employees, motivating them, boosting loyalty, and retaining talent (Greening and Turban, 2000) [38]. Third, it can reduce labor costs and boost sales revenue, thereby elevating enterprise value (Hamrouni et al., 2019) [28]. Finally, it enhances investors' understanding of an enterprise's non-financial information, fostering trust and fulfilling financing needs. For example, firms with strong ESG performance face lower compliance costs and adjust faster to negative events (Walker et al., 2014) [39]. Therefore, ESG-compliant firms are more likely to achieve sustainable growth, accruing intangible assets like brand goodwill.

However, extensive ESG investment does not necessarily lead to a steady increase in enterprise value (Wieczorek et al., 2021) [6]. Placing excessive emphasis on ESG performance can lead to resource constraints, leading to short-term losses for stockholders (Xue et al., 2022) [40]. While enterprises might transfer short-term losses to stockholders, this approach is not sustainable in the long run (Ohalehi, 2019) [41]. Excessive ESG costs may burden firms, leading to price hikes and consumer loss. Thus, while strong ESG can enhance enterprise value within limits, exceeding these limits can be detrimental. Therefore, the connection between ESG performance and enterprise value is unlikely to follow a linear pattern.

ESG responsibility fulfillment is an internal behavior of enterprises, while financing constraints are external factors influencing enterprises' conduct. Financing constraints impede stakeholder attention toward ESG, impairing enterprises' ability to enhance their value. First, in environments with strict financing constraints, enterprises mitigate risks through transparency. However, as ESG practices become more prevalent, their competitive advantage diminishes (Ma, 2019) [25].To mitigate financing risk and gain competitive advantage, some managers opt to enhance information transparency, minimize adverse selection risks, and cultivate investor trust through proactive ESG responsibility fulfillment (Alda, 2020) [42]. Over time, other enterprises follow suit. As ESG responsibility becomes

widespread, it ceases to be a unique competitive advantage. For early adopters, the cost of ESG responsibility fulfillment increases and may eventually become a financial burden.

Second, the market is more developed in areas of high financing constraints (Chen et al., 2012) [43]. And enterprises in developed markets are more likely to fulfill ESG responsibilities, fostering intense market competition. Enterprises continuously refine their ESG practices and accumulate management experience, facilitating knowledge spillover and further enhancing their value by fulfilling their ESG responsibilities. However, excessive investment in ESG responsibility can deplete resources and exacerbate its adverse impact on enterprise value, particularly in areas of high financing constraints.

Third, the level of stakeholder understanding regarding a company's ESG efforts is heavily influenced by the financing environment. In areas of high financing constraints, intense market competition often leads to greater information transparency and a more efficient market (Eccles et al., 2014) [44]. This transparency facilitates smoother communication between companies and stakeholders. Stakeholders can then react promptly to a company's ESG performance according to investment decisions and other actions (Gao et al., 2017) [45]. Conversely, low financing constraints are often associated with a lack of awareness and competition surrounding ESG practices. Stakeholders in such environments may have limited information about a company's ESG efforts, exacerbating information asymmetry and weakening the impact of ESG performance on enterprise value. Based on the above analysis, we propose hypothesis H1 to capture these dynamics.

**H1:** *The relationship between ESG responsibility fulfillment and enterprise value is non-linear, with financing constraints playing a significant moderating role in this relationship.*

### 2.2. The Heterogeneity of ESG Responsibility Fulfilment and Enterprise Value

Enterprises encounter diverse impacts when fulfilling their ESG responsibilities, influenced by factors such as the industry context and ownership structure. ESG responsibility entails a focus on environmental protection, aligned with the relevant policies and regulations in China. Enterprises can be categorized into polluting and non-polluting entities based on the industry characteristics. Moreover, they can be classified as state-owned or non-state-owned based on their ownership structure. Recognizing this diversity among enterprises is critical for accurately assessing the influence of ESG performance on enterprise value in China (Wang and Yang, 2022) [27].

ESG responsibility fulfillment can be a competitive advantage for some firms, positively influencing enterprise value. However, this influence depends on the industry characteristics and ownership structure. Non-polluting industries, with naturally lower environmental management costs (Zhang and Zhao, 2019) [46], can leverage strong ESG performance for a greater advantage. Yet resource constraints and market saturation due to knowledge spillover can impede further value creation.

In contrast, polluting enterprises face public pressure and additional costs for environmental compliance (Yin et al., 2022) [20]. In such industries, the competitive dynamics are weaker, and the knowledge spillover effect is less pronounced, resulting in differing impacts of ESG performance compared to in non-polluting sectors. Overall, the effect of ESG performance on enterprise value in non-polluting enterprises likely follows a non-linear pattern, shaped by the industry-specific characteristics and ownership structures.

Ownership structures play an important role (Xu et al., 2021) [47]. Non-state-owned enterprises are typically more agile in their internal controls compared to their state-owned counterparts. Consequently, their ESG performance tends to align better with the market demands. For non-state-owned enterprises, adherence to ESG responsibilities may stem from a focus on legal compliance. However, their focus on legal compliance might lead them to go beyond what is necessary, exceeding stakeholder expectations for value creation (Xu et al., 2022) [19].

Furthermore, the input–output dynamics of non-state-owned enterprises' ESG performance follow the principle of diminishing marginal returns in economics. Initially, fulfilling

ESG responsibilities adds positively to enterprise value. However, as these efforts intensify, the incremental value gain is likely to decrease. In contrast, state-owned enterprises, closely tied to government objectives, tend to prioritize government-led ESG initiatives rather than value enhancement.

To summarize, the impact of ESG performance on the value of non-state-owned enterprises likely follows a non-linear relationship. It may initially enhance value, but this effect diminishes as the ESG efforts intensify. In contrast, state-owned enterprises may engage in ESG compliance primarily for political reasons rather than value creation. Based on the above analysis, we propose hypothesis H2:

**H2:** *The effect of ESG responsibility fulfillment on enterprise value varies significantly based on the industry characteristics and ownership structure.*

### 2.3. Reducing Financing Cost Is the Primary Path for ESG Responsibility Fulfillment to Influence Enterprise Value

ESG compliance reduces financing costs by strengthening the ties between enterprises and stakeholders, increasing market transparency, and reducing investors' risk perceptions. Investors are more willing to provide unsecured debt to ESG-compliant enterprises, as seen in real estate trust (REIT) investments (Feng and Wu, 2021) [30]. Limkriangkrai et al. (2016) [48] studied the Australian market and found that ESG-compliant enterprises could pay fewer dividends after equity financing. Enhanced market transparency lowers the payoffs demanded by investors, reducing the cost of debt and equity financing and positively impacting enterprise value (Eliwa et al., 2021) [49]. Therefore, fulfilling ESG responsibilities can reduce financing costs.

The hypothesis of "information perfect symmetry" posits that enterprises have an optimal capital structure that maximizes their value (Hovakimian and Li, 2012) [50]. Changes in the capital structure directly impact enterprise value. However, real-world markets often lack perfect information, making the relationship between capital structure and value more complex. Debt issuers may closely monitor managers' debt financing activities, which can limit flexibility. In contrast, equity financing lacks strict repayment constraints. Equity financing may offer more freedom, magnifying its impact on enterprise value through financing costs. Therefore, hypothesis H3 is proposed:

**H3:** *ESG responsibility fulfillment primarily impacts enterprise value by lowering financing costs, with significant differences between the debt and equity financing routes.*

## 3. Research Design

### 3.1. Data Sources and Processing

This paper analyzes data from Shanghai and Shenzhen A-share listed enterprises in China from 2012 to 2020. More than 95% of Chinese public companies are listed on these two stock exchanges. The selection of these enterprises is based on two primary reasons. Firstly, the stringent listing process reflects the stability and substantial scale of these companies, ensuring comprehensive information on ESG responsibility fulfilment. Secondly, mainstream ESG rating agencies have data available only for listed enterprises from 2012 to 2020.

The data on ESG performance are derived by aggregating the ratings and scores from three institutions: Sino-Securities, Bloomberg, and Hexun. The Sino-Securities ESG data are retrieved from the Wind database, while the Bloomberg and Hexun data are manually collected. The other financial data are obtained from the CSMAR (China Stock Market & Accounting Research) database. The raw data undergo three processing steps. Firstly, enterprises labeled as ST and ST* are excluded due to financial troubles and/or potential delisting concerns. Secondly, those in the financial and real estate sectors, as well as those with gearing ratios exceeding 1, are also excluded due to their unique financial structures and risk profile. Thirdly, to mitigate the impact of outliers, all continuous variables undergo

1% and 99% tailing processing. These processing steps result in a final dataset comprising 9076 sample observations.

### 3.2. Variable Descriptions

#### 3.2.1. Dependent Variable

The existing studies typically utilize two main categories of indicators to measure enterprise value. One category comprises accounting indicators, including metrics like return on total assets (ROA), return on net assets (ROE), growth rate of asset size (ΔSize), and sales growth rate (ΔSales). The other category involves market indicators, with Tobin's Q being a prominent example. The users of ESG performance information include both internal personnel and numerous external stakeholders in the market. External stakeholders often assess an enterprise's ESG responsibilities based on its capital market value. Therefore, this paper opts to utilize Tobin's Q as a measure of enterprise value.

#### 3.2.2. Explanatory Variable

The current ESG performance ratings from mainstream institutions show certain variations. Sino-Securities updates its ratings widely, categorizing them into three levels (3A to A) across A, B, and C grades. Bloomberg offers segmented ESG scores, reflecting diverse perspectives and overall performance assessments. Meanwhile, Hexun's ratings consider multiple dimensions, like shareholders, employees, suppliers, products, and after-sales service. Table 1 presents the ESG compliance scores from these three organizations.

**Table 1.** Descriptive statistics of the three major ESG rating agencies.

| Institution | Sample Size | Mean | Standard Deviation | Minimum | Median | Maximum |
|---|---|---|---|---|---|---|
| Sino-Securities | 32,210 | 61.283 | 13.282 | 10.000 | 60.000 | 90.000 |
| Bloomberg | 9340 | 20.771 | 7.084 | 1.240 | 19.835 | 64.115 |
| Hexun | 31,984 | 23.267 | 15.134 | −18.450 | 21.540 | 90.87 |
| *WESG* | 9241 | 40.937 | 9.505 | 6.927 | 39.921 | 68.077 |

Data source: Wind et al. database.

Overall, except for Sino-Securities' ESG compliance data, Bloomberg's and Hexun's scores are relatively lower, indicating early ESG compliance stages in China. Specifically, Sino-Securities' ESG rating data boast a high overall score, while Bloomberg's data register a lower overall score, and Hexun's data exhibit the highest standard deviation. Recognizing these rating variations, this study follows the method outlined by Berg et al. (2022) [51] to create an ESG responsibility performance index (*WESG*).

We construct the ESG responsibility performance index (*WESG*) using a weighted average method. First, we standardize the ESG rating data from different institutions and then compute the weighted average based on predefined weights. This optimized approach ensures that the *WESG* indicator is comprehensive, comparable, and reasonably representative. The construction method for the *WESG* indicators is outlined as follows:

$$WESG_{it} = \sum ESG_{it} * I_i$$

$ESG_{it}$ indicates the ESG performance rating scores from each institution for a specific year $t$ of the $i$, while $I_i$ is the weight of each agency.

#### 3.2.3. Moderating and Mediating Variables

Three primary methods are used to assess the enterprise financing constraints: the single variable index method, sensitivity model method, and index method. The single variable index method is straightforward but lacks representativeness due to relying on a single factor. The sensitivity models are comprehensive but complex, labor-intensive, and prone to significant errors. Most index methods encompass KZ, SA, WW, and FC. The FC index, derived from logistic regression using standardized indicators like age and cash

dividend payout rate, serves as a crucial measure. Furthermore, the FC index consistently exceeds zero, with higher values signifying greater financing constraints for enterprises. Following Kuang (2011) [52] and Zhang et al. (2017) [53], this paper uses the FC index as the moderating variable to gauge enterprise financing constraints. Further, we test the moderating effect using KZ and SA indexes, both of which also demonstrate a significant moderating effect.

In terms of measuring the financing costs, this paper considers both debt and equity financing. For the cost of debt financing, the paper follows the approach of Wang and Yang (2022) [27] and selects the cost of enterprise debt financing (COD) as the mediating variable. In addition, following Li and Liu (2009) [54] and Zheng et al. (2013) [55], the ratio of financial expenses to liabilities is employed to calculate the cost of enterprise debt financing.

Meanwhile, the cost of equity financing (COE) is evaluated using the PEG model, as suggested by Mao et al. (2012) [56] and Yang et al.(2015) [57]. Specifically, the cost of equity financing (COE) is calculated as follows:

$$COE = \sqrt{\frac{EPS_{t+2} - EPS_{t+1}}{P_t}}$$

EPS, which stands for Earnings Per Share, follows Mao and Botosan's approach, utilizing analysts' forecasts of the earnings per share as the earnings per share for the sample companies in each period. $EPS_{t+2}$ and $EPS_{t+2}$ represent the forecast earnings per share by analysts in periods $t + 2$, and $t + 1$. $Pt$ represents the price per share of the enterprise at the end of the $t$ period.

### 3.2.4. Control Variables

Drawing from the characteristics of Shanghai and Shenzhen A-share listed enterprises, this study integrates the findings from Wang and Yang (2022) [27] and Xu et al. (2022) [19]. Consequently, we incorporate enterprise size (Size), growth capacity (Growth), fixed asset ratio (Fixed), and the proportion of the largest shareholder (Top1), along with year and industry dummy variables as the control variables, into our analysis. Size is measured using a natural logarithm for the total assets, while Growth is determined according to the operating income growth rate. Fixed represents the sum of net fixed assets and inventory divided by the enterprise assets. Top1 denotes the percentage of shareholders holding the most shares in the company. To mitigate the influence of time on our empirical findings, year dummy variables are included in the regression model. Recognizing significant variations among listed enterprises across different industries in Shanghai and Shenzhen A-shares, industry dummy variables are also integrated as control variables. Table 2 presents the relevant variables used in this study.

The main variables' descriptive statistics are shown in Table 3. Tobin's Q, the dependent variable, ranges from 0.855 to 11.980, with a standard deviation of 1.763, indicating significant variability across enterprises. *WESG*, the explanatory variable, ranges from 6.927 to 68.08, with a median of 39.960, suggesting generally modest values. This reflects the early stage of ESG development in China, indicating room for improvement in ESG responsibility fulfillment. The moderating variable, FC, ranges from 0 to 1, with an average of 0.310, showing generally low financing constraints among the sampled enterprises.

The mediating variable, debt financing (COD), ranges from −0.188 to 0.070. Negative COD values do not indicate negative actual debt financing costs but rather serve as a negative agent indicator for COD. Equity financing ranges from 0.012 to 0.262, averaging at 0.095, which is lower than the median of 0.120. This indicates a left-biased trend in equity financing among the sampled enterprises. The sample size for equity financing is 3550, significantly smaller than the other sample sizes, due to the stringent condition requiring a positive difference between the analysts' forecast earnings per share for two consecutive years for meaningful calculation.

**Table 2.** Description of the main variables.

| Variable Type | Variable Name | Variable Definition |
|---|---|---|
| Dependent variable | Tobin's Q | Market value divided by total assets at the end of the period |
| Explanatory variable | *WESG* | Weighted ESG responsibility fulfillment optimization indicator |
| Moderating variable | FC | Standardized enterprise age and cash dividend payout ratio |
| Mediating variables | COD | Ratio of financial expenses to liabilities |
| | COE | Analysts' forecast of the difference in earnings per share for two consecutive years divided by the difference between the closing share price and the opening price |
| Control variables | Size | Natural logarithm of total assets of an enterprise |
| | Growth | The growth rate of revenue |
| | Fixed | The sum of net fixed assets and net inventory divided by total assets |
| | Top1 | The largest shareholder's ownership ratio |
| | Year | 2012–2020 |
| | Ind | Industry classification guidelines for listed enterprises according to the SFC 2012 guidelines |

**Table 3.** Descriptive statistics of the main variables.

| Variable | Sample Size | Mean | Standard Deviation | Minimum | Median | Maximum |
|---|---|---|---|---|---|---|
| Q | 9076 | 2.253 | 1.763 | 0.855 | 2.552 | 11.980 |
| *WESG* | 9076 | 40.960 | 9.494 | 6.927 | 39.960 | 68.080 |
| FC | 8865 | 0.310 | 0.245 | 0.006 | 0.490 | 0.942 |
| COD | 9076 | 0.011 | 0.032 | −0.188 | 0.029 | 0.070 |
| COE | 3550 | 0.095 | 0.050 | 0.012 | 0.120 | 0.262 |
| Size | 9076 | 23.030 | 1.274 | 19.570 | 23.820 | 26.020 |
| Growth | 9076 | 0.159 | 0.401 | −0.561 | 0.237 | 2.856 |
| Fixed | 9076 | 0.242 | 0.176 | 0.003 | 0.348 | 0.700 |
| Top1 | 9076 | 36.930 | 15.920 | 8.790 | 48.800 | 75.050 |

### 3.3. Empirical Model

To examine the non-linear relationship between ESG responsibility fulfillment and enterprise value, as deduced from the theoretical analysis, we incorporate the quadratic term of enterprise ESG responsibility fulfillment ($WESG^2$) into the fixed-effect model. This inclusion allows us to assess the impact of ESG performance on enterprise value. Thus, we formulate the following model:

$$Q = \beta_0 + \beta_1 WESG + \beta_2 WESG^2 + \beta_3 Control + YearEffect + IndEffect + \varepsilon_{i,t} \quad (1)$$

To examine the moderating impact of financing constraints on the relationship between ESG performance and enterprise value, we introduce an interaction term for the relationship between financing constraints (FC) and explanatory variables (*WESG*) into the model (2). Testing hypothesis H1 relies on the significance of this term. If it is significant, this would suggest the presence of a moderating effect of financing constraints on the relationship between ESG responsibility fulfillment and firm value.

$$Q = \beta_0 + \beta_1 WESG + \beta_2 WESG^2 + \beta_3 WESG \times FC + \beta_4 Control + YearEffect + IndEffect + \varepsilon_{i,t} \quad (2)$$

To analyze this mechanism, the cost of debt financing (COD) and cost of equity financing (COE) of enterprises act as mediating variables and are evaluated using a three-step model. The framework for the mediation test is outlined as follows:

$$Q = \beta_0 + \beta_1 WESG + \beta_2 WESG^2 + \beta_3 Control + YearEffect + IndEffect + \varepsilon_{i,t} \quad (3)$$

$$COD(COE) = \beta_0 + \beta_1 WESG + \beta_2 Control + YearEffect + IndEffect + \varepsilon_{i,t} \quad (4)$$

$$Q = \beta_0 + \beta_1 WESG + \beta_2 WESG^2 + \beta_3 COD(COE) + \beta_4 Control + YearEffect + IndEffect + \varepsilon_{i,t} \quad (5)$$

$\beta_0$ represents a constant term; $\beta_{Control}$ denotes the control variables; $YearEffect$ represents the year fixed effect, $IndEffect$ represents the industry fixed effect; $i$ represents the year; $t$ denotes the sample enterprises; $\varepsilon_{i,t}$ represents the error terms.

Furthermore, considering the potential endogeneity between ESG performance and enterprise value, we utilize analyst forecast bias and industry averages of ESG performance for two-stage least-squares estimation, effectively addressing the potential endogenous problem (Fatemi et al., 2017; Wang and Yang, 2022) [27,58].

## 4. Empirical Results and Analysis

### 4.1. Analysis of the Results of the Benchmark Regression

Table 4 displays the regression results examining the relationship between ESG performance and enterprise value. The coefficient estimate for $WESG$ is statistically significant and positive, indicating that a higher ESG performance is positively associated with enterprise value. Conversely, the coefficient estimate for $WESG^2$ is significantly negative, suggesting a shift in the effect of ESG performance from positive to negative with greater resources allocated to ESG responsibility fulfillment.

**Table 4.** Baseline regression and instrumental variable regression results.

| Variable | (1) | (2) | (3) First Stage (ESG) | (4) First Stage (ESG²) | (5) Second Stage |
|---|---|---|---|---|---|
| | Q | Q | ESG | ESG² | Q |
| WESG | 0.057 *** | 0.075 *** | | | 1.031 *** |
| | (5.21) | (6.87) | | | (5.30) |
| WESG² | −0.001 *** | −0.001 *** | | | −0.012 *** |
| | (−4.50) | (−5.42) | | | (−5.39) |
| WESG × FC | | −0.025 *** | | | |
| | | (−9.84) | | | |
| ERROR | | | −0.147 *** | −11.250 *** | |
| | | | (−8.96) | (−8.06) | |
| mESG | | | 1.020 *** | 90.635 *** | |
| | | | (49.58) | (51.81) | |
| Size | −0.746 *** | −0.853 *** | 1.392 *** | 112.373 *** | −0.684 *** |
| | (−23.41) | (−23.07) | (18.97) | (18.02) | (−22.33) |
| Growth | 0.308 *** | 0.328 *** | −0.898 *** | −81.553 *** | 0.490 *** |
| | (10.61) | (11.26) | (−3.96) | (−4.23) | (8.93) |
| Fixed | −0.651 *** | −0.512 *** | −2.37 *** | −188.271 *** | −1.346 *** |
| | (−3.85) | (−3.00) | (−4.57) | (−4.27) | (−9.98) |
| Top1 | 0.000 | 0.002 | −0.003 | −0.496 | 0.000 |
| | (0.24) | (0.78) | (−0.53) | (−1.00) | (0.14) |
| cons | 17.894 *** | 19.927 *** | −31.533 *** | −4417.133 *** | −2.996 |
| | (16.56) | (17.44) | (−16.55) | (−27.27) | (−0.85) |
| N | 9076 | 8865 | 8495 | 8495 | 8495 |
| R² | 0.263 | 0.267 | Instrumental Variable Correlation Test: | | |
| Ind | Yes | Yes | Unidentifiable test: *p* value = 0.000 | | |
| Year | Yes | Yes | Weak instrumental variable test: F value = 33.469 | | |

*** indicates significance at 1% level, and the values in parentheses denote Z-values.

The regression analysis identifies the mean ESG performance score at the inflection point as 28.5. Below this threshold, there is a positive association between ESG performance and enterprise value. However, above this threshold, ESG performance starts to negatively impact enterprise value. In essence, while engaging in ESG responsibility ful-

fillment is beneficial for enterprises, excessive efforts in this area can lead to a decline in enterprise value.

In summary, the empirical regression results support a non-linear, inverted U-shaped relationship between ESG performance and enterprise value, supporting hypothesis H1. To account for the influence of external factors such as financing constraints, we augment the baseline regression model by introducing an interaction term for the relationship between financing constraints and ESG responsibility fulfillment (*WESG* × FC) into the baseline regression model. The regression results in Column (2) indicate significant coefficients for both the linear and quadratic terms, revealing a blend of positive and negative influences. This reaffirms the inverted U-shaped relationship between ESG performance and enterprise value.

Moreover, the coefficient for the interaction term for the relationship between financing constraints and ESG performance is statistically significant at the 1% level, highlighting the moderating role of financing constraints. Specifically, a higher numeric value for an enterprise's financing constraint indicates more severe constraints. Upon incorporating the interaction term, the coefficient for the linear term of ESG responsibility performance increases from 0.057 to 0.075. This suggests that under financing constraints, the positive impact of ESG performance on enterprise value becomes more pronounced. Thus, hypothesis H1 is affirmed.

### 4.2. Discussion and Treatment of the Endogenous Problem

There is a potential issue of endogeneity between ESG performance and enterprise value, as successful firms are more likely to fulfill ESG responsibilities to enhance their reputation. To address this endogeneity concern, we incorporate instrumental variables into our model. Following the approach of Fatemi et al. (2017) [58] and Wang and Yang (2022) [27], we perform a two-stage least-squares analysis using analyst forecast bias (ERROR) and the industry average for ESG responsibility fulfillment (mESG) as instrumental variables. Columns (3)–(5) in Table 4 present the results of the instrumental variable regression. Columns (3) and (4) display the first-stage regression outcomes, providing fitted values for the endogenous explanatory variable ESG responsibility fulfillment and its squared term. Column (5) presents the second-stage results of the instrumental variable regression.

The findings show a notably negative coefficient for analyst forecast bias (FERROR) among the instrumental variables, signaling an adverse relationship between analyst forecast bias and ESG performance. This implies that higher ESG performance might help alleviate information asymmetry and lessen the variability in the analyst forecasts.

Moreover, a higher ESG performance within the industry in which an enterprise operates signifies an overall dedication to ESG responsibilities within that industry. As a result, the industry mean of ESG responsibility fulfillment should exhibit a positive correlation with the efforts of individual enterprises in fulfilling their ESG responsibilities.

Combining the results from the instrumental variable regression, there is a positive coefficient for mESG, indicating a favorable impact. Furthermore, the second-stage analysis confirms that the linear coefficient for ESG responsibility fulfillment remains significantly positive, while the quadratic coefficient is significantly negative. This reaffirms the inverted U-shaped relationship between ESG performance and enterprise value, even after addressing the endogeneity issue.

Moreover, we conduct unidentifiable and weak instrumental variable tests, presented in Columns (3)–(5) in the table. The *p*-value of the unidentifiable test is 0, signifying statistical significance. In addition, the F-value of the weak instrumental variable test is 33.469, surpassing the threshold of 10, indicating the validity of the instrumental variables used in this study. As there are two endogenous variables in the model, matching the number of instrumental variables, there is no need for over-identification tests (Staiger and Stock, 1994) [59].

*4.3. Robustness Tests*

4.3.1. Replacement of the Dependent Variable and Independent Variables

To ensure the robustness and reliability of our baseline regression results, we conduct robustness tests by experimenting with alternative variables. To measure enterprise value, we substitute the market value of a non-tradable equity with its net assets divided by its total assets at the period's end. This substitution results in the variable Q1. The outcomes of this substitution into the benchmark regression model are presented in Column (1) of Table 5. Despite the small quadratic coefficient, its negative value indicates an inverted U-shaped relationship between ESG performance and enterprise value.

**Table 5.** Robustness test.

| Variable | (1) | (2) | (3) q25 | (4) q50 | (5) q75 | (6) q90 |
|---|---|---|---|---|---|---|
| | Q1 | Q | Q | Q | Q | Q |
| *WESG* | 0.032 *** | | −0.013 *** | 0.003 | 0.027 | 0.081 *** |
| | (3.66) | | (−2.98) | (0.41) | (1.61) | (3.36) |
| *WESG²* | −0.000 *** | | 0.000 *** | −0.000 | −0.000 | −0.001 *** |
| | (−2.63) | | (3.35) | (−0.00) | (−1.43) | (−3.08) |
| *WESG1* | | 0.077 *** | | | | |
| | | (4.18) | | | | |
| *WESG1²* | | −0.001 *** | | | | |
| | | (−3.34) | | | | |
| Control | Yes | Yes | Yes | Yes | Yes | Yes |
| N | 9076 | 4793 | 9076 | 9076 | 9076 | 9076 |
| R² | 0.209 | 0.289 | 0.116 | 0.142 | 0.166 | 0.191 |

*** indicates significance at 1% level, and the values in parentheses are t-values.

Furthermore, the explanatory variables are enhanced by incorporating data from Runling's global ESG responsibility rating, along with ratings from four other mainstream institutions. These rating scores are weighted to construct the ESG responsibility performance indicators (*WESG1*). The regression result incorporating these additional variables is presented in Column (2) of Table 5. Notably, the inverted U-shaped relationship between ESG performance and enterprise value persists, confirming the robustness of the benchmark regression results.

4.3.2. Quantile Regression

Given that the panel data fixed-effect model primarily examines the average-level impact of ESG performance on enterprise value, it fails to capture the variations across different levels of enterprise value. To address this concern, we conduct regressions using quartiles—specifically, the 0.25, 0.5, 0.75, and 0.9 quartiles—to assess the effect of ESG performance on enterprise value. The results, displayed in Columns 3–6 of Table 5, indicate significance for both the 25% and 90% quartiles, suggesting the effectiveness of the non-linear relationship at low and high levels of enterprise value.

Analyzing the coefficients' sign direction, we find that at the 25% level, the one-term coefficient is significantly negative, while the quadratic coefficient is significantly positive. This indicates that the impact of ESG performance on enterprise value initially declines and then increases when the enterprise value is low. In contrast, at the 90% level of enterprise value, the one-term coefficient is significantly positive, while the quadratic coefficient is significantly negative. This suggests that at high levels of enterprise value, ESG responsibility fulfillment initially boosts enterprise value before dampening it.

In summary, our analysis highlights the significant non-linear relationship between ESG performance and enterprise value, particularly evident at low and high levels of enterprise value.

### 4.4. Heterogeneity Analysis

According to the theory of resource conservation, enterprises tend to avoid activities that deplete their resources once they reach a certain resource threshold (Grant, 1999) [60]. The costs and benefits of ESG compliance vary across industries. For instance, non-polluting enterprises can utilize their resources more efficiently, as they do not incur additional costs for pollution management. This implies that the initial impact of ESG compliance may be more pronounced for non-polluting enterprises. To investigate this further, we classify listed enterprises based on industry using the SFC 2012 Industry Classification Guidelines for Listed Enterprises. Subsequently, we conduct regression analysis for enterprises within different industry categories.

The results presented in Columns (1) and (2) of Table 6 show a significantly positive coefficient for the linear term and a significantly negative coefficient for the quadratic term in non-polluting industries. This suggests that ESG performance initially enhances enterprise value, but once the ESG score reaches 48.5, its positive impact diminishes, indicating an inverted U-shaped relationship. In contrast, for enterprises in polluting industries, the linear coefficient is significantly negative, with no significant nonlinear relationship observed.

**Table 6.** Heterogeneity test of industries and ownership structures.

| Variable | (1) Non-Polluting | (2) Polluting | (3) Non-State-Owned | (4) State-Owned |
|---|---|---|---|---|
| | Q | Q | Q | Q |
| *WESG* | 0.097 *** | −0.033 * | 0.069 *** | 0.010 |
| | (6.36) | (−1.78) | (3.46) | (0.75) |
| *WESG*$^2$ | −0.001 *** | 0.000 | −0.001 *** | −0.000 |
| | (−5.86) | (1.62) | (−3.02) | (−0.78) |
| Control | Yes | Yes | Yes | Yes |
| N | 5826 | 3250 | 4348 | 4728 |
| R$^2$ | 0.129 | 0.089 | 0.139 | 0.080 |

* indicates significance at 10% level, *** indicates significance at 1% level, and the values in parentheses are t-values.

State-owned enterprises, which control over 60% of China's market capitalization, typically respond to government-led initiatives on ESG responsibility earlier and more forcefully than non-state-owned enterprises. On the other hand, non-state-owned enterprises possess greater flexibility in their internal controls and can adjust their business processes more quickly. To examine this phenomenon further, we classify the sample enterprises into state-owned and non-state-owned categories based on their ownership structures. Subsequent group regression analysis yields the results presented in Columns (3) and (4) of Table 6. The regression coefficients reaffirm the presence of an inverted U-shaped relationship for non-state-owned enterprises.

In summary, the nonlinear effect of ESG performance on enterprise value is more pronounced for non-polluting or non-state-owned enterprises, thus confirming hypothesis H2.

### 4.5. Further Discussion: Mechanism Analysis

In the current economic landscape, transitioning from virtual to real economic growth requires rigorous financial oversight, especially concerning information disclosure, such as financing channel and capital utilization on the part of listed enterprises. Chinese listed companies often grapple with limited access to financing and high financing costs. To explore how ESG responsibility fulfillment influences enterprise value, we adopt a three-step method to analyze the mediating role of financing costs.

Columns (1) and (2) of Table 7 depict the impact of debt financing costs as an intermediary variable on enterprise value. The results in Column (1) reveal a significantly negative regression coefficient for ESG responsibility fulfillment in terms of debt financing costs, suggesting that fulfilling ESG responsibilities can mitigate the cost of debt financing for

enterprises. However, the coefficient for debt financing costs in Column (2) lacks statistical significance, indicating that a pathway involving debt financing costs is not supported.

**Table 7.** The test of the mediating effect.

| Variable | (1) | (2) | (3) | (4) |
|---|---|---|---|---|
| | COD | Q | COE | Q |
| *WESG* | −0.000 *** | 0.056 *** | −0.001 *** | 0.059 *** |
| | (−3.37) | (5.08) | (−3.94) | (2.85) |
| *WESG*$^2$ | | −0.001 *** | | −0.001 *** |
| | | (−4.38) | | (−2.59) |
| COD | | −0.742 | | |
| | | (−1.44) | | |
| COE | | | | −1.543 *** |
| | | | | (−3.67) |
| Control | Yes | Yes | Yes | Yes |
| N | 9076 | 9076 | 3550 | 3550 |
| R$^2$ | 0.095 | 0.263 | 0.076 | 0.278 |

*** indicates significance at 1% level, and the values in parentheses are t-values.

Moving on to Columns (3) and (4), we examine the influence of equity financing costs as an intermediary variable on enterprise value. Column (3) demonstrates that ESG responsibility fulfillment can indeed reduce the cost of equity financing, as evidenced by the significantly negative coefficient for equity financing costs in Column (4). This finding suggests that ESG performance can impact enterprise value by lowering the cost of equity financing.

In summary, while the pathway of ESG responsibility fulfillment's impact on enterprise value through debt financing costs remains inconclusive, it appears to primarily influence enterprise value by reducing equity financing costs. Consequently, ESG responsibility fulfillment emerges as a critical indicator for evaluating enterprise value. Moreover, equity financing serves as a signaling mechanism that influences enterprise behavior. The confirmation of the heterogeneity of the impact of ESG responsibility fulfillment on enterprise value through the debt and equity financing pathways underscores the significance of the latter, particularly highlighted by the effectiveness of the equity financing cost pathway. Thus, our analysis provides empirical support for hypothesis H3.

## 5. Conclusions

ESG practices have grown globally, with Chinese enterprises rapidly closing the gap with their European and American counterparts. By 2021, over half of Chinese enterprises had issued ESG reports, a trend catalyzed by China's ambitious climate targets announced in 2020. This context underscores the urgent need to evaluate how ESG performance influences the values of listed Chinese companies. To meet this imperative and building upon prior studies like Wang et al. (2022) [27], we introduce a novel ESG responsibility metric using a regulated intermediary model. This approach considers the industry characteristics and ownership structures. Further, our methodology addresses potential endogeneity issues using instrumental variables and two-stage least-squares methods, ensuring the reliability of our findings.

Our results highlight the non-linear effect of ESG performance on enterprise value, especially concerning financial constraints. We reveal an inverted U-shaped relationship, indicating that improving ESG performance initially boosts enterprise value, yet excessive investment in ESG beyond a threshold can yield negative effects. Financial constraints moderate this relationship, especially in non-polluting and non-state-owned enterprises. In addition, enhanced ESG performance increases enterprise value by lowering equity financing costs, although it does not show a significant correlation with debt financing.

Our study strengthens the reliability of our findings by addressing endogeneity concerns using instrumental variables and two-stage least-squares methods. This approach

aligns with the research by Li et al. (2022) [23] and Feng and Wu (2021) [30]. The implications for corporate decision-making and resource optimization are substantial, particularly within the Chinese market context. We also emphasize the heterogeneity in the relationship between ESG performance and enterprise value across different enterprise categories.

For the research community, our nonlinear model and regulated intermediary model provide new analytical tools for studying the complex relationship between ESG performance and enterprise value. These methodologies can be applied to investigating a wide range of sustainability-related issues. For instance, researchers can use our approach to analyze topics such as how employee relations affect innovation and productivity (Shah, et al., 2022) [61], the impact of tax practices on investment decisions and financial transparency (Chughtai, et al., 2021) [62], consumer purchasing decisions and brand loyalty (Moslehpour, et al., 2021) [63], government incentives for corporate investment in renewable energy and environmentally friendly technologies (Hashmi, et al., 2021) [64], and the impact of various emission reduction strategies on corporate costs, competitiveness, and market performance (Rjoub, et al., 2021) [65], as well as improvements in patient care quality and employee satisfaction in nursing leadership (Wang, et al., 2022) [66]. Interested readers can refer to Wong et al. (2020) [67] and Wong (2020) [68] for more information on these topics.

Based on our findings, we recommend optimizing resource allocation based on the inverted U-shaped relationship, ensuring that social responsibility efforts are in line with their economic benefits for sustainable development and value maximization. The impact of ESG practices on enterprise value is accentuated by high financial constraints, indicating the need for strategies that reduce financing costs and enhance efficiency. Tailoring ESG strategies to suit industry characteristics and ownership structures can improve both value creation and societal impact. Companies can benefit from evaluating and adjusting their capital structure, such as transitioning from high-cost debt to equity or low-cost debt. Strengthening investor relations and leveraging ESG performance for improved market visibility can also contribute to reduced financing costs. Furthermore, the use of diversified financial instruments like green bonds and investments aligned with Sustainable Development Goals (SDGs) can further decrease costs while boosting enterprise value. These strategic initiatives aligning financial objectives with sustainable practices can yield positive outcomes for both businesses and society.

While providing important insights, our study has its limitations. Firstly, our sample comprises A-share listed companies in China, limiting the generalizability of our conclusions to global markets and diverse legal and cultural contexts. Secondly, our focus on the ESG performance–enterprise value relationship does not include other factors like technological innovation, market dynamics, and macroeconomic conditions that influence enterprise value. Thirdly, our analysis lacks depth in exploring the impacts of individual ESG dimensions, particularly the environmental, social, and governance aspects, on enterprise value. Moreover, as noted by Dang et al., 2021 [69], this model may not fully address the endogeneity issue, which can stem from unobservable firm and CEO characteristics. There is also the potential for fraudulent reporting.

To overcome these limitations, future studies should consider assessing the long-term impacts on corporate reputation and exploring broader sustainability-related issues using non-linear modeling. These research efforts could extend to diverse regions and industries, including developed and developing markets, various company sizes, and types. It is crucial to delve deeper into the individual ESG dimensions, examining their independent and collective effects on enterprise value. For instance, the research could investigate how improved environmental practices lead to cost reductions or how social responsibility activities boost brand value, consumer loyalty, and reputation. Furthermore, investigating potential "greenwashing" practices is necessary, utilizing text analysis and interviews to understand how companies shape their ESG image using language and disclosure strategies aligned with actual business practices and ethical standards. Evaluating the long-term impact of ESG practices on corporate reputation through tracking studies and market

research can also provide valuable insights into public perceptions and their influence on corporate reputation and value assessments.

**Author Contributions:** Conceptualization, X.S.; Methodology, G.S.; Software, X.S.; Validation, X.S., G.S. and Z.Y.; Formal analysis, X.S. and G.S.; Investigation, X.S. and Z.Y.; Data curation, G.S.; Writing— original draft, G.S. and Z.Y.; Project administration, X.S. All authors have read and agreed to the published version of the manuscript.

**Funding:** The first author is grateful for the support of Henan Province Higher Education Teaching Reform Research and Practice Project "Innovation and Practice of New Business Talent Training Model in Local Universities for Digital Economy" (2024SJGLX139), the 2022 Henan Philosophy and Social Science Planning Project "Research on the Government Empowerment Mechanism for Improving the Ability of Poverty Alleviation Farmers to Build Assets" (grant number: 2022BJJ089), the 2022 Henan Soft Science Research Project "Research on the Path of Improving Henan's Total Factor Productivity Under High-Quality Economic Development" (grant number: 222400410087), the 2021 Henan New Liberal Arts Research and Reform Practice Project "Innovation and Practice Research on the New Business Talent Training System of Local Colleges and Universities Under the Empowerment of Information Technology" (grant number: JG [2021] No. 175), and the Postgraduate Education Reform and Quality Improvement Project of Henan Province (YJS2022JD30).

**Institutional Review Board Statement:** Not applicable.

**Informed Consent Statement:** Not applicable.

**Data Availability Statement:** The ESG related data of Huazheng in this article is sourced from the Wind database (https://www.wind.com.cn/portal/zh/WFC/index.html); ESG data from Bloomberg and Hexun.com was collected through their websites, while additional data was sourced from the CSMAR database (https://data.csmar.com).

**Conflicts of Interest:** The authors declare no conflict of interest.

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
