# Peer review of "ESG Performance and Enterprise Value in China: A Novel Approach via a Regulated Intermediary Model"

_sustainability, doi:10.3390/su16083247_

Round 1

Reviewer 1 Report

Comments and Suggestions for Authors

The paper entitled "ESG Performance and Enterprise Value in China: A Novel Approach through the Regulated Intermediary Model" is the result of the authors' research regarding the introduction of a novel ESG responsibility performance metric through the Regulated Intermediary Model to delve into this relationship within the Chinese context. As a result, the authors reveal an inverted U-shaped relationship between ESG performance and enterprise value, with financing constraints having a significant moderating effect.

Overall, I am satisfied with the way the analysis was conducted.

I have only a few suggestions for authors:

1. Avoiding footnotes.

2. A standardization of the tables;in the sense of using the same character size, the same font.

3. To ensure (especially for general readers) that all the elements included in the tables are described (as an example, what does "*" or "***" mean in table 6 and the same remark for the other tables)

4. A proofreading of the manuscript is required to correct small technical editing errors, punctuation marks and others like these: for example, line 495 mentions table 10, which does not exist in the manuscript or reference no. 24 which seems incomplete (". ial Analysis. 84, 102382")

Thank you!

Author Response

The paper entitled "ESG Performance and Enterprise Value in China: A Novel Approach through the Regulated Intermediary Model" is the result of the authors' research regarding the introduction of a novel ESG responsibility performance metric through the Regulated Intermediary Model to delve into this relationship within the Chinese context. As a result, the authors reveal an inverted U-shaped relationship between ESG performance and enterprise value, with financing constraints having a significant moderating effect.

Overall, I am satisfied with the way the analysis was conducted.

I have only a few suggestions for authors:

  1. Avoiding footnotes.

Response: Thank you for your suggestion. We have relocated the footnote from the introduction section to the references section at the end (References [3] and [5], respectively), provided explanations for the footnotes in the study design section, and highlighted the corresponding changes in red in both sections.

  1. A standardization of the tables in the sense of using the same character size, the same font.

Response: Thank you for the suggestion. Following your advice, we have uniformly adjusted the format of all tables in this article. Unfortunately, some font size and type changes after we upload the manuscript.

  1. To ensure (especially for general readers) that all the elements included in the tables are described (as an example, what does "*" or "***" mean in table 6 and the same remark for the other tables)

Response: We appreciate this comment. We have made changes to the tables. In all tables, *, **, and *** denote significance at the 10%, 5%, and 1% levels, respectively. The values in parentheses in Table 4 represent Z-values, while the values in brackets in Tables 5 to 7 represent t-values. We have added this description below the tables, highlighted in red for clarity.

  1. A proofreading of the manuscript is required to correct small technical editing errors, punctuation marks and others like these: for example, line 495 mentions table 10, which does not exist in the manuscript or reference no. 24 which seems incomplete (". ial Analysis. 84, 102382")

Response: Thank you for the careful reading. We have indeed made a typographical error on line 495, showing Table 10. We have corrected this error and highlighted it in red for clarity. Additionally, regarding the incomplete presentation of article 24 in the references, we have rectified this error and also highlighted it in red (now listed as the 36th reference).

Reviewer 2 Report

Comments and Suggestions for Authors

The paper analyzed a current and debated topic regarding the relationship between ESG and firm value for Chinese listed firms, using the Regulated Intermediary Model, for a period of nine years (2012-2020).

I like the paper.

Strenghts:

-        The title of the paper is clear and adequate;

-        The abstract describes appropriately the content of the paper and it presents the research objectives and the results;

-        The content of the paper (internal organization) is well structured and has clarity;

-        The paper includes the appropriate literature (references) for the chosen topic.

-        The methodology apllied is correct;

-        The contribution to the literature is well highlighted from Introduction section.

Recommendations:

1) The Conclusion section is too sketchy and can be improved by discusing more the implication of the findings.

2) Reference 24, line 642 has to be corrected.

Author Response

The paper analyzed a current and debated topic regarding the relationship between ESG and firm value for Chinese listed firms, using the Regulated Intermediary Model, for a period of nine years (2012-2020).

I like the paper.

Strengths:

The title of the paper is clear and adequate;

The abstract describes appropriately the content of the paper and it presents the research objectives and the results;

The content of the paper (internal organization) is well structured and has clarity;

The paper includes the appropriate literature (references) for the chosen topic.

The methodology applied is correct;

The contribution to the literature is well highlighted from Introduction section.

Recommendations:

1) The Conclusion section is too sketchy and can be improved by discussing more the implication of the findings.

Response:

The feedback is greatly appreciated, leading to a restructuring of the conclusion section in response. Initially structured into five paragraphs covering results summary, suggestions, ESG framework importance, specific model applications, and study limitations, it was found to lack depth.

The revised conclusion is not made into seven paragraphs to improve clarity and depth. The seven paragraphs covered the importance of this research, key findings, methodological advancements, major academic contributions, recommendations, study limitations, and future directions.

The revised conclusion now offers a more comprehensive and structured overview of the study's findings and implications. It condenses the key conclusions derived from the analysis, highlighting an inverted U-shaped relationship between ESG performance and enterprise value, especially under varying financial constraints. The recommendation section emphasizes optimizing resource allocation based on this relationship to balance social responsibility and economic benefits effectively. Additionally, the conclusion delves into the implications for businesses, society, and scientific research, emphasizing the significance of ESG practices in contemporary business environments and the need for future research to address broader contexts and dimensions beyond the scope of the current study.

Overall, the revised conclusion bridges the gap between research findings and practical applications, providing a roadmap for future investigations and encouraging a more nuanced understanding of ESG's role in corporate strategies, societal impact, and scientific advancements. It serves as a potential guide for researchers, businesses, and policymakers aiming to navigate the complexities of ESG integration.

2) Reference 24, line 642 has to be corrected.

Response: Thank you for your suggestion, and we apologize for making such a mistake. We have carefully proofread and corrected the reference in the original 24th article (now the 36th article). Additionally, we have checked other references one by one to avoid similar errors.

Reviewer 3 Report

Comments and Suggestions for Authors

Hello,

 To improve the article please:

- to argue and explain the representativeness of the sample of 9,076 sample observations;

- to present in conclusion a comparative approaches with similar studies;

- to present in more details future research directions.

Please find my specific comments below:

The study presents ESG responsibility performance metric through the Regulated Intermediary Model.

The objectives of the study are explicit. The conclusions are well formulated and correspond to the content of the scientific analyses carried out but are not placed in a broader scientific context.

The subject has been studied by other authors. The references to other studies are sufficient and clarify very well the developments in the field.

The methodology is current and correct.

Few problematisations are presented that can support future studies.

The authors mention the limitations of the research.

Good luck!

Comments on the Quality of English Language

Minor editing of English language required!

Author Response

Hello,

 To improve the article please:

- to argue and explain the representativeness of the sample of 9,076 sample observations;

Response: We apologize for not explaining the sample selection process more clearly.

Our sample consists of all regular enterprises listed on the Shanghai Stock Exchange and Shenzhen Stock Exchange in China from 2012 to 2020. We chose 2012 as the starting year due to data availability for ESG ratings of Chinese enterprises, and we set the data cutoff at 2020 to avoid COVID-19 impacts. Our focus on listed Chinese enterprises excludes small ones prone to short-term bankruptcy, with most listed on the Shanghai and Shenzhen Stock Exchanges. We excluded ST and ST* labeled samples due to delisting risks and firms from finance, real estate sectors, or with debt-to-asset ratios exceeding 1 due to significant operational differences and insolvency risks. This approach aligns with prior literature. We have also consulted literature that similarly excludes these two sectors. For example, Zheng, Zhigang, et al. (2023) [1] excluded enterprises from the financial sector, and Gao J., Chu D., Lian Y., et al. (2021) [2] omitted data from both the financial and real estate sectors. Lastly, To address outliers in our sample, we utilized Stata 17.0 software. Specifically, we applied the Winsor2 command for a two-tailed winsorization across all data points. This technique involved identifying values below the 1st percentile and above the 99th percentile as outliers and replacing them with the values at the 1st and 99th percentiles, respectively. Through these processes, we obtained data from 9,076 enterprises and deemed it to be representative to a certain extent.

  • Zheng, Zhigang, et al. "Does corporate ESG create value? New evidence from M&As in China." Pacific-Basin Finance Journal 77 (2023): 101916.
  • Gao J., Chu D., Lian Y., et al. Can ESG performance improve enterprise investment efficiency? [J]. Securities Market Herald, 2021(11):24-34+72.

- to present in conclusion a comparative approaches with similar studies;

Response: Thank you for your valuable suggestion. We have expanded the conclusion section to seven paragraphs by comparing it with other relevant studies, highlighting how our research complements existing literature and demonstrating how our new methodology can be used in other subjects. Our study identifies an inverted U-shaped relationship between ESG performance and enterprise value, with financial constraints moderating this relationship significantly. Unlike previous studies, we introduce a novel approach, the Regulated Intermediary Model, to measure ESG responsibility performance in the Chinese market context. Our findings provide fresh evidence on the non-linear impact of ESG practices on enterprise value under varying financial constraints. Addressing endogeneity concerns, our study employs instrumental variables and two-stage least squares methods, enhancing the reliability of our findings, echoing Li et al. (2022). Our research contributes to shaping corporate strategic decisions and resource allocation, emphasizing the role of ESG performance in reducing financing costs, in line with Feng and Wu (2021). Furthermore, we highlight heterogeneity in the relationship between ESG performance and enterprise value within non-polluting and non-state-owned enterprises.

- to present in more details future research directions.

Response:

Thank you for your suggestion. We have expanded the section on future research directions in paragraph 4 of the conclusion section to provide detailed insights into how our study can guide future research. Our discussion now outlines how our research methodology can be applied to other sustainability-related issues and proposes specific recommendations for future studies.

We have also strengthened the section for future research in paragraph 7 of the conclusion section. Future research could explore broader regions and industries, analyzing the independent and collective impacts of ESG dimensions on enterprise value. To address potential 'greenwashing,' studies could use text analysis and interviews to assess companies' ESG image alignment with ethical standards. The non-linear model we employed offers a new analytical tool for studying ESG performance and its impacts, applicable to exploring other sustainability issues. For example, research could analyze how employee relations influence innovation and productivity or how tax practices affect financial transparency.

Reviewer 4 Report

Comments and Suggestions for Authors

Main Comments and Suggestions

The paper does not do a good job of discussing the implications of the findings. Why do we care? How to find the optimal ESG to maximize EV? You should clarify the contributions of the paper which are not elaborated well in the current paper. You can talk about the following contributions: What insights can you provide based on your findings? Do they push forward our understanding? What should we do with your research? Do you have any suggestions to improve the current regulation or practice? Adding the above discussion and extending your literature review may help you make more contributions and position your contributions better.

The paper seems to claim causality but the IVs are not truly exogenous. The analyst forecast bias (ERROR) and the industry average of ESG responsibility fulfillment (mESG) as instrumental variables are problematic. The endogeneity problem can be driven by unobservable firm and CEO characteristics you need to discuss. Here are my suggestions: first, you can claim that Error is mostly driven by analyst talent, which is largely exogenous to valuation, per Dang et al 2021. Journal of Corporate Finance. Second, you can argue industry ESG, though driven by industry features, can be exogenous when you use industry fixed effects to control for these features.

You need more literature to support the finding of inverted U shape, specifically for the downward part. The literature suggests that Chinese listed firms may use CSR/ESG to cover up fraud as in Li et al 2024. Corporate Social Responsibility and Goodwill Impairment: Charitable Donations of Chinese Listed Companies. You can provide some guidance on how firms can achieve the maximum valuation through adjusting ESG (something like optimal capital structure).

Minor Comments and Suggestions

Try to avoid long sentences and vague words. Use short, precise, and concise sentences and be more straightforward. The last section of the conclusion should summarize all your findings, their implications to researchers and practitioners, future direction for research, limitations of the current study, etc. You need to seriously proofread the paper and extend and update your references.

In conclusion, I would like to thank the authors for a very interesting, unique and potentially important paper. Hope these comments and suggestions can help further their study.

Comments on the Quality of English Language

need thorough proofreading

Author Response

The paper does not do a good job of discussing the implications of the findings. Why do we care? How to find the optimal ESG to maximize EV? You should clarify the contributions of the paper which are not elaborated well in the current paper. You can talk about the following contributions: What insights can you provide based on your findings? Do they push forward our understanding? What should we do with your research? Do you have any suggestions to improve the current regulation or practice? Adding the above discussion and extending your literature review may help you make more contributions and position your contributions better.

Response:

Thank you for your valuable feedback. We have expanded the conclusions section to delve deeper into the significance of our research findings for our readers. Specifically, we emphasized the rapid growth of ESG practices, the impact of China's climate targets, and the crucial need to understand the ESG performance-enterprise value relationship. We have also added three paragraphs in the conclusion section to highlight the policy contribution, our contribution to other research subjects, and future research directions.

Here are some specific ones:

1. Understanding the Non-linear Relationship: Our research highlights the non-linear relationship between ESG performance and enterprise value, indicating that while enhancing ESG performance can initially strengthen enterprise value, there exists a threshold beyond which excessive investment in ESG may lead to diminishing returns or even negative impacts on enterprise value. This nuanced understanding is crucial for companies aiming to strike a balance between social responsibility and economic benefits.

2. Moderating Role of Financial Constraints: Financial constraints play a significant moderating role in the relationship between ESG performance and enterprise value, particularly for non-polluting and non-state-owned enterprises. This suggests that companies facing financial challenges can benefit from improving their ESG performance to reduce financing costs and optimize financing efficiency.

3. Contribution for the Research Community: Our nonlinear model and Regulated Mediation Model provide new analytical tools to future explore the complex relationship between ESG performance and enterprise value and investigate a wide range of sustainability-related issues.

4. Recommendations for Companies: Companies are advised to optimize their resource allocation based on the inverted U-shaped relationship between ESG responsibility fulfillment and enterprise value. It is crucial for companies to ensure that pursuing social responsibility does not unduly sacrifice enterprise value, thereby achieving sustainable development and maximizing value.

5. Tailored ESG Strategies: The finding that different types of enterprises exhibit varied patterns in the relationship between ESG performance and enterprise value implies that companies can develop more precise and effective ESG strategies based on their industry characteristics and ownership structures. This tailored approach can enhance both enterprise value and societal impact.

6. Policy Implications: Our research suggests potential avenues for improving current regulations or practices. For instance, regulators could consider providing incentives for companies to improve their ESG performance, especially those facing financial constraints. Moreover, policy frameworks could be designed to encourage companies to disclose more transparent and accurate information regarding their ESG practices and their impacts on enterprise value. Meanwhile, state-owned-enterprises will not respond to financial constraints as much as non-state-owned-enterprises regarding ESG responsibility fulfillment.

We have incorporated your suggestions into the introduction and the conclusion sections.

The paper seems to claim causality but the IVs are not truly exogenous. The analyst forecast bias (ERROR) and the industry average of ESG responsibility fulfillment (mESG) as instrumental variables are problematic. The endogeneity problem can be driven by unobservable firm and CEO characteristics you need to discuss. Here are my suggestions: first, you can claim that Error is mostly driven by analyst talent, which is largely exogenous to valuation, per Dang et al 2021. Journal of Corporate Finance. Second, you can argue industry ESG, though driven by industry features, can be exogenous when you use industry fixed effects to control for these features.

Response: We apologize for the lack of clarity in our explanation for using industry averages of analyst forecast errors (ERROR) and ESG responsibility fulfillment (mESG) as instrumental variables to address endogeneity issues. Analysts' forecasts of a company's future earnings may be influenced by the company's ESG performance. Generally, companies with better ESG performance tend to receive more accurate analyst forecasts (Fatemi et al., 2017), reflecting greater transparency and comprehensive information disclosure (as supported by literature)..

However, analysts' forecast bias is unlikely to directly impact a company's value, making it a suitable instrumental variable to isolate the independent effect of ESG responsibility fulfillment on corporate value. The industry average level of ESG responsibility fulfillment reflects the sector's overall attention and practice regarding ESG. This variable correlates with individual companies' ESG behavior, influenced by industry standards and peer pressure. Yet, industry average ESG levels are unlikely to directly affect individual companies' value, making them effective instrumental variables (Wang and Yang, 2022) for identifying the impact of ESG responsibility fulfillment on corporate value.

We do recognize that the use of the instrumental variable may not tease out all reserve causality. Therefore, we have added reference #74, included discussions in paragraph 6 of the conclusion section, and acknowledged this potential limitation.

  • Fatemi, A., Glaum, M., and Kaiser, S. (2018). ESG performance and firm value: The moderating role of disclosure. Global finance journal. 38, 45-64. doi:10.1016/j.gfj.2017.03.001.
  • Wang B., and Yang M. (2022). A study on the mechanism of ESG performance on corporate value—empirical evidence from A-share Listed companies in China. Soft Science. 36(06):78-84. doi:10.13956/j.ss.1001-8409.2022.06.11.

You need more literature to support the finding of inverted U shape, specifically for the downward part. The literature suggests that Chinese listed firms may use CSR/ESG to cover up fraud as in Li et al 2024. Corporate Social Responsibility and Goodwill Impairment: Charitable Donations of Chinese Listed Companies. You can provide some guidance on how firms can achieve the maximum valuation through adjusting ESG (something like optimal capital structure).

Response: Your suggestions are valuable; we acknowledge the importance of enhancing our research with extensive literature support. We synthesized research relevant to the inverted U-shaped relationship between ESG responsibility fulfillment and corporate value, focusing on the descending segment.

Regarding Li et al. 2024, we could only find one related study titled "ESG Performance and Corporate Fraud," highlighting how "fulfilling ESG responsibilities inhibits fraudulent behavior in Chinese companies" rather than covering it up, as you mentioned. Several other studies support the positive impact of ESG responsibility fulfillment on companies, not the descending part of the inverted U-shaped relationship.

We appreciate the insights provided regarding "Corporate Social Responsibility and Goodwill Impairment: Charitable Donations of Chinese Listed Companies," as goodwill impairment may occur if the company's value declines. However, upon reviewing the relevant literature, we found that studies using Chinese listed companies as samples have discovered that companies with more long-term donations tend to report goodwill impairment promptly, while those with excessive short-term donations are more likely to delay goodwill impairment. This is because the motivation for short-term donations is not only to cover up the delay in goodwill impairment but also to provide an insurance-like protective mechanism when announcing delayed impairment (Li, Zhichuan Frank, Zhaozhe Lu, and Jian Wang, 2023). Alternatively, a company's Corporate Social Responsibility (CSR) can prevent goodwill impairment (Golden, Joanna, Li Sun, and Joseph H. Zhang, 2018).

This study elaborates that the fulfillment of ESG responsibilities by Chinese companies has inhibited fraudulent behavior, rather than the "CSR/ESG to cover up fraud " that you referred to. Moreover, some of the references cited in this study also support the positive impact of ESG responsibility fulfillment on companies, rather than supporting the descending part of the inverted U-shaped relationship (He, Feng, Hanyu Du, and Bo Yu, 2022).

Your suggestions on exploring whether ESG responsibility fulfillment masks fraudulent behavior and its impact on corporate goodwill align with future research directions outlined in our conclusion. While your suggestions seem to align with the real-world, relevant literature directly supporting these cases is lacking. To address this issue, we have included future research directions in our conclusion to highlight the potential for fraudulent reporting and explore potential "greenwashing".

Our literature review has found that the aforementioned topics do not support the descending part of the inverted U-shaped relationship. Therefore, we have argued for the descending part of the inverted U-shaped relationship from the perspective of the resource consumption and increased costs associated with the fulfillment of ESG responsibilities by Chinese listed companies. Of course, your suggestions are indeed pertinent and closely aligned with the actual situation. There are indeed relevant news reports in China, such as the case of Sanyuan Food (stock code: 600429), whose subsidiary, Beijing Shougang Livestock Development Co., Ltd., and its branches received multiple environmental protection penalty notices due to their arbitrary discharge of wastewater through means of evading regulation, causing environmental pollution (source: https://www.mrjjxw.com/articles/2022-04-16/2222745.html). Such companies claim to "fulfill ESG responsibilities" while engaging in activities that pollute the environment.

We cannot rule out the possibility that they use ESG for greenwashing and defraud stakeholders, thereby reducing the company's value, but this actual situation lacks the support of related research. In conclusion, your suggestions are very pertinent, and we agree, but we have not found literature that matches this actual situation. Once again, we must express our gratitude to you, as your suggestions have pointed our research in the right direction. Consequently, we have included the direction for future research in the fourth paragraph of the conclusion section of our article: to explore whether the fulfillment of ESG responsibilities by Chinese listed companies masks fraudulent behavior and whether it has had a detrimental impact on corporate goodwill, etc. The modifications we have made are as follows:

“……It is crucial to delve deeper into individual ESG dimensions, examining their independent and collective effects on enterprise value. For instance, research could investigate how improved environmental practices lead to cost reductions or how social responsibility activities boost brand value, consumer loyalty, and reputation. Furthermore, investigating potential "greenwashing" practices is necessary, utilizing text analysis and interviews to understand how companies shape their ESG image through language and disclosure strategies aligned with actual business practices and ethical standards. Evaluating the long-term impact of ESG practices on corporate reputation through tracking studies and market research can also provide valuable insights into public perceptions and their influence on corporate reputation and value assessments……”

Your suggestions are indeed very valuable! In accordance with your recommendations, we have included several proposals in the conclusion section, such as how companies can adjust their capital structure to facilitate the optimal impact of ESG responsibility fulfillment on corporate value, in order to explore how businesses can find the optimal level of ESG practices. With the addition of this section, our conclusions are now more specific, vivid, and practically meaningful compared to before! We have attached the revised conclusion section below for your review, and we have also highlighted the changes in red within the original text:

“……Tailoring ESG strategies to suit industry characteristics and ownership structures can improve both value creation and societal impact. Companies can benefit from evaluating and adjusting their capital structure, such as transitioning from high-cost debt to equity or low-cost debt. Strengthening investor relations and leveraging ESG performance for improved market visibility can also contribute to reduced financing costs. Furthermore, the use of diversified financial instruments like green bonds and investments aligned with Sustainable Development Goals (SDGs) can further decrease costs while boosting enterprise value. These strategic initiatives align financial objectives with sustainable practices, leading to positive outcomes for businesses and society alike……”

  • Li, Dengjia, et al. "ESG performance and Corporate Fraud." Finance Research Letters (2024): 105212.
  • He, Feng, Hanyu Du, and Bo Yu. "Corporate ESG performance and manager misconduct: Evidence from China." International Review of Financial Analysis 82 (2022): 102201.
  • Li, Zhichuan Frank, Zhaozhe Lu, and Jian Wang. "Corporate Social Responsibility and Goodwill Impairment: Evidence from Charitable Donations of Chinese Listed Companies." Available at SSRN 4337571 (2023).
  • Golden, Joanna, Li Sun, and Joseph H. Zhang. "Corporate social responsibility and goodwill impairment." Accounting and the Public Interest 18.1 (2018): 1-28.

Try to avoid long sentences and vague words. Use short, precise, and concise sentences and be more straightforward. The last section of the conclusion should summarize all your findings, their implications to researchers and practitioners, future direction for research, limitations of the current study, etc. You need to seriously proofread the paper and extend and update your references.

Response: Thank you for your valuable feedback. We have improved the clarity and precision of our paper by avoiding ambiguous language and long sentences. Our revisions include proofreading to correct grammatical and spelling errors, as well as replacing complex sentences with simpler ones for better readability.

Specifically, we focused on restructuring sentences to ensure clear and precise articulation of each argument. We conducted a detailed vocabulary examination to eliminate ambiguity and ensure that each word carries a distinct meaning. Following this, we conducted a comprehensive proofreading of the entire manuscript to ensure grammatical and spelling accuracy. Complex sentences were replaced with concise and straightforward ones to enhance overall clarity and readability.

Furthermore, we have updated our references to include more relevant and recent sources, aligning with your suggestion to enhance the credibility and relevance of our work. In the conclusion section, we have added paragraphs, summarized our key findings, discussed their implications for both researchers and practitioners, outlined potential avenues for future research, and acknowledged the limitations of our study. These revisions are highlighted in red for your review.

Round 2

Reviewer 2 Report

Comments and Suggestions for Authors

I recommend the paper for publication.

Author Response

We have further revised the introduction and the conclusion sections. Thank you very much! 

Reviewer 4 Report

Comments and Suggestions for Authors

Improved. You need to add all the literature in your response file to the paper and update reference list. Also consider my suggestions regarding endogeneity problem: first, you can claim that Error is mostly driven by analyst talent, which is largely exogenous to valuation, per Dang et al 2021. Journal of Corporate Finance. Second, you can argue industry ESG, though driven by industry features, can be exogenous when you use industry fixed effects to control for these features.

Comments on the Quality of English Language

need more work

Author Response

Dear Reviewer,

We would like to express our gratitude for the opportunity to revise our manuscript titled “ESG Performance and Enterprise Value in China: A Novel Approach via the Regulated Intermediary Model”. We appreciate the thorough feedback provided by the reviewers and have made significant improvements to our manuscript based on their suggestions.

Firstly, We apologize that the reference and the additional texts were missed in our last revision, even though we discussed them in our response letter. In this revised version, we have rectified these omissions and further revised the abstract, introduction, and conclusion sections. The changes have been highlighted in red for easy identification. Our focus has been on enhancing readability while maintaining the coherence and depth of our research.

Specifically, we have addressed your instructions by shortening lengthy sentences to improve clarity and flow throughout the manuscript. Additionally, we have strengthened the section on contributions to the academic community by expanding the fifth paragraph in the conclusion, emphasizing the novel insights our study offers to the field.

Furthermore, we have added Dang et al 2021. Journal of Corporate Finance to our reference and expanded discussions in the limitations section (sixth paragraph in the conclusion) to highlight potential endogeneity issues. Additionally, we have added a future research direction regarding the long-term impact of ESG practices on corporate reputation, which can provide valuable insights into public perceptions and their influence on corporate value assessments.

Thank you for the opportunity to revise our manuscript.